# Histological, Physiological and Transcriptomic Analysis Reveal Gibberellin-Induced Axillary Meristem Formation in Garlic (*Allium sativum*)

**DOI:** 10.3390/plants9080970

**Published:** 2020-07-31

**Authors:** Hongjiu Liu, Yanbin Wen, Mingming Cui, Xiaofang Qi, Rui Deng, Jingcao Gao, Zhihui Cheng

**Affiliations:** College of Horticulture, Northwest A&F University, Yangling 712100, China; liured9@nwsuaf.edu.cn (H.L.); freejustice@nwsuaf.edu.cn (Y.W.); cuimm@nwafu.edu.cn (M.C.); Qi-Xiaofang@nwafu.edu.cn (X.Q.); durant@nwsuaf.edu.cn (R.D.); gaojingcao@nwsuaf.edu.cn (J.G.)

**Keywords:** *Allium sativum*, gibberellin, axillary meristem, sucrose, transcriptome, *AsGA20ox*

## Abstract

The number of cloves in a garlic bulb is controlled by axillary meristem differentiation, which directly determines the propagation efficiency. Our previous study showed that injecting garlic plants with gibberellins (GA_3_) solution significantly increased clove number per bulb. However, the physiological and molecular mechanism of GA-induced axillary bud formation is still unknown. Herein, dynamic changes in histology, phytohormones, sugars and related genes expression at 2, 4, 8, 16 and 32 days after treatment (DAT) were investigated. Histological results indicated two stages (axillary meristem initiation and dormancy) were in the period of 0–30 days after GA_3_ treatment. Application of GA_3_ caused a significant increase of GA_3_ and GA_4_, and the downregulation of *AsGA20ox* expression. Furthermore, the change trends in zeatin riboside (ZR) and soluble sugar were the same, in which a high level of ZR at 2 DAT and high content of soluble sugar, glucose and fructose at 4 DAT were recorded, and a low level of ZR and soluble sugar arose at 16 and 32 DAT. Overall, injection of GA_3_ firstly caused the downregulation of *AsGA20ox*, a significant increase in the level of ZR and abscisic acid (ABA), and the upregulation of *AsCYP735* and *AsAHK* to activate axillary meristem initiation. Low level of ZR and soluble sugar and a high level of sucrose maintained axillary meristem dormancy.

## 1. Introduction

Shoot branching, originating from an axillary bud, is an important determinant of plant architecture and significantly influences crop yield [1]. Axillary bud development is involved in two stages: axillary meristem initiation in the leaf axil and axillary bud outgrowth or dormancy [2]. For decades, the role of auxins, cytokinins (CKs), strigolactones (SLs) and brassinosteroid (BR) in shoot branching was reported, revealing a complex network of signals that combine to regulate an axillary meristem into a branch [3]. In addition, sugars and their signaling networks also played a critical role at the early stages of axillary bud outgrowth [4,5]. Currently, a relatively clear picture of phytohormones and sugars regulating axillary bud development was established in *Arabidopsis thaliana* [6,7], tomato [8], rice [9,10], barley [10], apple [11] and rose [12].

Garlic, the second most important *Allium* crop after the bulb onion, is cultivated and consumed worldwide for its flavor and medicinal value [13]. The garlic bulb, normally consisting of 8–15 cloves, is not only the main production organ, but also the propagation organ because most garlic cultivars are sterile [14]. The clove includes a bud, a storage leaf and a protective leaf, which is equated with tiller in rice and branch in woody plants [13]. Furthermore, gibberellins (GAs), as indispensable stimulators of plant growth [15], are applied to induce lateral bud outgrowth in tomato [16], branches in cherry trees [17] and *Jatropha curcas* [18], and tiller in Welsh Onion [19]. Our previous study showed injection of GA_3_ promoted axillary meristem formation and increased clove number per bulb [20], which is a better tool to enhance propagation efficiency and bulb yield of garlic [21]. Nevertheless, the physiological mechanism of GA-induced axillary meristem formation of garlic, especially the role of plant hormones and sugars, is still unknown.

Genetic studies in *Arabidopsis thaliana* [22], rice [23] and pea [24] have shown that axillary meristem initiation is regulated by several transcription factor-encoding genes, such as *CUP-SHAPED COTYLEDON* (*CUC*), *LATERAL SUPPRESSOR* (*LAS*), *REGULATOR OF AXILLARY MERISTEMS* (*RAX*), and *REVOLUTA* (*REV*) in *Arabidopsis*. However, it has been reported that *FLOWERING LOCUS T* (*FT*) genes regulated axillary meristem formation and clove number in garlic [25]. As *Allium sativum* is a nonmodel plant; its genomic information is still scare, except for the development of molecular markers [26,27,28], which restricts our understanding of the molecular mechanism of axillary meristem development of garlic. With the development of sequencing technologies, RNA sequencing (RNA-seq) independent of genetic background has been developed [29]. Currently, RNA-seq has been utilized to elucidate the response of garlic clove (shoot apical meristem) dormancy to storage temperature [30,31] and organ-specific profiling of gene expression in fertile garlic [32]. Carbohydrate genes and *FT* participation in the regulation of meristem termination and bulbing in garlic was reported [30]. Even though the understanding of the molecular mechanism of axillary meristem initiation and garlic bulbing has gradually improved, the regulatory genes and network of axillary meristem development in garlic have never been reported. Hence, we examined in this study the global gene expression changes under water and GA_3_ treatment using Illumina RNA-Seq technology.

GAs are a large group of tetracyclic diterpenes regulating many aspects of plant growth and development [33], such as promoting stem growth in sunflower [34], enhancing secondary xylem development in carrot [35] and inducing fruit-set in tomato [36]. GAs biosynthesis is catalyzed by six pivotal enzymes, involving ent-copalyl diphosphate synthase (CPS), ent-kaurene synthase (KS), ent-kaurene oxidase (KO), enthaurenoic acid oxidase (KAO), GA 20-oxidase (GA20ox), GA 3-oxidase (GA3ox) [33,37]. *GA20ox* function as a key player in producing bioactive GAs in plants and is cloned from many plant species [38,39,40,41]. Importantly, a high level of GAs in leaf axils by ectopically expressing *GA20ox2* dramatically inhibited axillary meristem initiation in *Arabidopsis* [42]. However, it is unknown whether *GA20ox* expression plays a critical role in GA-induced axillary bud formation of garlic.

Our previous studies indicated GAs, CKs, IAA (indole-3-acetic acid), sugars and soluble protein participated in the process of GA-induced axillary bud formation of garlic [20,21,43]. In this study, we further investigated the changes of phytohormone levels, sugars content and the expression levels of key genes during the process of GA-induced axillary meristem formation of garlic. Our results not only provide a new and relative clear network of hormone and sugars metabolism for GA-induced axillary meristem formation of garlic, but also enrich our knowledge of improving garlic propagation efficiency via exogenous GA_3_.

## 2. Results

### 2.1. Injection of GA_3_ Promotes Axillary Meristem Formation of Garlic

Axillary meristem appeared after 30 days of GA_3_ treatment; on the contrary, there was no axillary meristem in the control at the same time (Figure 1A,B). At 90 days after treatment (DAT), significantly more axillary meristem (young axillary bud) formation took place, while no axillary meristem was yet recorded in control (Figure 1C,D). Mature axillary bud arose after 120 days of GA_3_ treatment, which was regarded as the first time of axillary bud formation under GA_3_ treatment (Figure 1E,F). There were axillary buds in both control and GA_3_ treatment at 150 DAT (Figure 1G,H). Obviously, a large number of axillary buds were found in GA_3_ treatment, which was regarded as the second time of axillary bud formation under GA_3_ treatment.

After harvest, the clove number and clove arrangement were evaluated on the basis of Figure 1I. Clove number and bulb structure were dramatically affected by GA_3_ treatment, with several cloves arranged around main bulb in GA_3_ treatment (Figure 1J). Importantly, the mean number of cloves per bulb was significantly increased by GA_3_ treatment (20.42), as compared to control (11.19). GA_3_ treatment also significantly increased whorl number per bulb (Figure 1K).

### 2.2. Impact of Injection of GA_3_ on Plant Hormone Level in Garlic Stems

The 0–30 days after GA_3_ treatment was a critical period in which axillary meristem (bud) formed and went into dormant status in this experiment (Figure 1 and Appendix A), based on our previous study [20,21]. Hence, plant hormone levels in stems in control and GA_3_ treatment were investigated at 2, 4, 8, 16 and 32 DAT. ZR (zeatin riboside) level was significantly increased after 2 days of GA_3_ treatment and sharply decreased after 8, 16 and 32 days of GA_3_ treatment, as compared to control (Figure 2A). By contrast, the change trend of IAA is opposite to ZR. The values of IAA of GA_3_-treated plants significantly decreased by 98.06% at 8 DAT, increased by 32.75% at 16 DAT, compared to control (Figure 2B). ABA (abscisic acid) level in GA_3_-treated plant (35.72 mg g^−1^ fresh weight (FW)) was significantly higher than control at 2 DAT (26.92 mg g^−1^ FW; Figure 2C). Furthermore, injection of GA_3_ significantly increased GA_3_ level at 2, 4, 8, 16 and 32 DAT (25.50, 21.34, 21.34, 20.58 and 7.58 mg g^−1^ FW, respectively), compared to control (4.37, 3.88, 3.95, 3.27 and 1.29 mg g^−1^ FW, respectively; Figure 2D). GA_4_ level was significantly increased by GA_3_ treatment at 2, 4 and 8 DAT (Figure 2E).

### 2.3. Effect of GA_3_ Treatment on Sugars and Soluble Protein Content in Garlic Stems

The changes of soluble sugar, soluble protein, sucrose, glucose and fructose in the stems of GA_3_-treated plants were detected at 2, 4, 8, 16 and 32 DAT. The value of soluble sugar was sharply increased by 34.28% after 4 days of GA_3_ treatment, but significantly reduced by 17.98% and 29.66% after 16 and 32 days of GA_3_ treatment, respectively (Figure 3A). However, injection of GA_3_ only caused an increase in soluble protein content at 8, 16 and 32 DAT (Figure 3B). Sucrose content was significantly increased by GA_3_ treatment at 8 and 32 DAT (11.45 and 16.24 mg g^−1^ FW, respectively), compared to control (9.36 and 14.10 mg g^−1^ FW, respectively; Figure 3C). Glucose content increased on the average of 179% and 110% at 4 and 32 day after GA_3_ treatment, respectively (Figure 3D). Fructose content was increased by 24.19% at 4 DAT (compare GA_3_ treatment versus control; Figure 3E).

### 2.4. Transcriptome Analysis

A total of 503,769,826 raw reads were obtained from the transcriptome sequencing of control (water treatment) and GA_3_ treatment at 2 DAT using the Illumina platform. After removing low-quality reads, 494,115,858 clean reads were mapped to the maize reference genome (https://ftp.ncbi.nim.nilh.gov/genomes/GCF_000005005.2_B_73_ErfGen_v4_genomic.fna.gz) by HISAT2. Under normal conditions, if the high quality of reference genome was chosen properly, and the relevant experiments were not contaminated, the percentage of total mapped reads would be higher than 70%. The mapping results showed that the lowest mapping rate is higher than 75%, suggesting the high quality of our transcriptome data (Table 1).

A total of 159 differentially expressed genes (DEGs) were detected in the comparison of GA_3_ treatment versus water treatment at 2 DAT (Figure 4A and Appendix A). To survey the potential functions of genes differentially expressed between control and GA_3_ treatment, GO (Gene Ontology) and KEGG (Kyoto Encyclopedia of Genes and Genomes) enrichments were performed for function classification. GO analysis showed that 83 DEGs were enriched in 5 GO terms, in which 6 and 77 genes were enriched in biological processes and molecular function, respectively (Figure 4B). Of the 74 genes, 37 and 37 were classified to be associated with heterocyclic compound binding and organic cyclic compound binding, respectively (Figure 4B). Four genes into phenylpropanoid biosynthesis, three genes into ubiquinone and other terpenoid-quinone biosynthesis, three genes into starch and sucrose metabolism and one gene into diterpenoid biosynthesis were detected in KEGG analyses (Figure 4C).

Furthermore, we listed 12 important unigenes that relate to plant hormone and sugars metabolism in a sample of control and GA_3_ treatment (Table 2). Among these 12 unigenes, 4 were DEGs, which encode gibberellin 20 oxidase 2, beta-glucosidase 31, beta-glucosidase 25 and 1,4-alpha-glucan-branching enzyme 3. As for the other 8 unigenes, they were mainly involved in the GA signal pathway, cytokinin biosynthesis, cytokinin signaling pathway, auxin signaling pathway, sucrose metabolism and meristem development. Therefore, the changes of GAs biosynthesis and sugars metabolism initially happened, which is considered as the first step for GA-induced axillary meristem formation of garlic.

### 2.5. Dynamic Expression of AsGA20ox, AsAUX, AsCYP735, AsAHK, AsBGLU31 and AsINV

Based on the results of transcriptome analysis, *Allium sativum gibberellin (GA) 20-oxidase* homolog (*AsGA20ox*; Cluster-32430.71391), *Allium sativum auxin-induced protein* homolog (*AsAUX*; Cluster-32430.172148), *Allium sativum cytokinin hydroxylase* homolog (*AsCYP735*; Cluster-15446.0), *Allium sativum histidine kinase* homolog (*AsAHK*; Cluster-32430.98625), *Allium sativum beta-glucosidase 31* homolog (*AsBGLU31*; Cluster-32430.190012), *Allium sativum invertase* homolog (*AsINV*; Cluster-32430.143582) were screened for the RT-qPCR experiment. In addition, the coding sequence (CDS; 1146 bp) of *AsGA20ox* was amplified from the transcriptome data (Cluster-32430.71391; Appendix A). The CDS (1929 bp) of *AsINV* was amplified from the transcriptome data (Cluster-32430.143582; Appendix A). The transcriptome data indicated expression of *AsGA20ox* and *AsBGLU31* were significantly downregulated, and expression of *AsCYP735* and *AsAHK* were upregulated in the stem after 2 days of GA_3_ treatment as compared to control (water treatment). No significant difference was recorded in expression of *AsAUX* and *AsINV* in the stems of GA_3_ treatment and control at 2 DAT (Table 2). As expected, the results of RT-qPCR were similar to transcriptome data (Figure 5 and Appendix A). Expression of *AsGA20ox* was significantly lower in stems under GA_3_ treatment compared to control at 2, 4, 8, 16 and 32 DAT (Figure 5A). In addition, expression of *AsAUX* was sharply downregulated by GA_3_ treatment at 8, 16 and 32 DAT (Figure 5B). By contrast, expression of *AsCYP735* and *AsAHK* were significantly higher at 2 and 4 DAT (GA_3_ treatment versus control; Figure 5C,D), but RNA-seq results showed there was no significant difference in expression of *AsCYP735* and *AsAHK* at 2 DAT (Table 2). In terms of sugar metabolism, expression of *AsINV* was 5-fold higher in stems after 32 days of GA_3_ treatment versus control (Figure 5E). On the contrary, expression of *AsBGLU31* was significantly downregulated in the GA_3_ treatment group at 2, 4 and 8 DAT (Figure 5F).

### 2.6. Principal Component Analysis (PCA)

PCA analysis was performed to show the trends, patterns and differences in the phytohormone level, sugars content, soluble protein content and expression level of key genes in stems of GA_3_-treated plants and water-treated (control) plants at 2, 4, 8, 16 and 32 DAT. PCA analysis revealed that four highest ranking principal components accounted for 85.69% of total variance. PC1 and PC2 accounted for 37.18% and 22.10% of the total variance, respectively, which showed the “GA_3_ (2d)”, “GA_3_ (4d)”, “GA_3_(8d)”, “GA_3_ (16d)”, “GA_3_ (32d)” and “Control (32d)” being distinct from “Control (2d)”, “Control (4d)”, “Control (8d)” and “Control (16d)” (Figure 6A). Further analysis showed that PC1 loadings were negative for GA_3_ (−0.53), GA_4_ (−0.66) and ABA (−0.68), but were positive for other parameters, showing an inverse relationship between “GA_3_, GA_4_, ABA” and other parameters (Figure 6B). The corresponding loadings were negative for ZR (−0.14), *AsGA20ox* (−0.56), *AsAUX* (−0.48) and *AsBGLU31* (−0.19), but were positive for other parameters, accounting for 22.10% of the variance in PC2 (Figure 6B).

## 3. Discussion

Previous studies suggested GAs play a negative role in the regulation of shoot branching in various species [44,45,46]. Recently, stimulation of GAs in axillary bud development was reported in cherry tree [17], *Jatropha curcas* [18] and Welsh onion [19]. Our previous study showed injection of GA_3_ four times (at 40, 50, 60 and 70 days after planting) promoted axillary bud formation and increased clove number per bulb (control and GA_3_ treatment: 12 and 24, respectively) in cv. G064 [20]. Similarly, injection of GA_3_ once (at 40 days after planting) also induced axillary bud formation and increased clove number per bulb (control and GA_3_ treatment: 12 and 20, respectively), which indicated the effects of injection of GA_3_ once and four times on clove number per bulb were the same. In addition, the process of GA-induced axillary bud formation of garlic in this study was the same as our previous results [21]. Because there was no significant difference in the size of axillary meristem among 16 DAT, 30 DAT and 60 DAT (Figure 1 and Appendix A), it was possible that 30 and 60 DAT was in winter with a relatively low temperature to induce axillary meristem dormancy. Hence, two stages, axillary meristem initiation and inhibition of axillary meristem growth (or axillary meristem dormancy), happened during 0–30 days after GA_3_ treatment. Meanwhile, our findings confirmed again that axillary bud formed twice in GA_3_-treated plants (Figure 7B), which caused a sharp increase in clove number per bulb to enhance garlic propagation efficiency [20].

A negative correlation between GAs level and axillary meristem formation was found in various plant species [47,48,49]. Our group also reported that seed cloves soaked in GA_3_ solution led to axillary bud formation with a low level of endogenous GA_3_ in the stem [43]. However, high levels of GAs in stems of GA_3_-treated plants were found in this study, thereby we suspected that application mode (injection) might result in residual GA_3_ in the stem. Furthermore, *GA20ox*, encoding a GA-biosynthesis enzyme, played various roles in plant development [50]. Ectopically expressing *GA20ox* led to an increase in GAs at the leaf axil to inhibit axillary meristem initiation in *Arabidopsis* [42]. Our findings showed significant downregulation of *AsGA20ox* in the stem, suggesting endogenous GAs level at the leaf axil is possibly reduced for activating the process of GA-induced axillary meristem formation. Nowadays our group is investigating how GAs metabolize and its downstream genes affect axillary meristem formation of garlic.

Wang et al. [51] found that cytokinin accumulation in the leaf axil partially rescued axillary meristem initiation-deficient mutants. Additionally, axillary bud outgrowth occurred with an increase of cytokinin levels in *Allium sativum* [43] and *Lupinus angustifolius* [52]. As expected, our results showed a high level of ZR and upregulation of *AsCYP735* (encoding a key enzyme of CKs biosynthesis) that were detected 2 days after GA_3_ treatment, which could participate in the promotion of axillary meristem initiation, as indicated in previous studies [5,53]. High expression of *AsAHK* arose after 2 and 4 days of GA_3_ treatment, indicating that the cytokinin signal pathway might positively regulate axillary meristem initiation [54]. Meanwhile, ABA, as a stress hormone, was increased by GA_3_ treatment at 2 DAT to induce axillary meristem formation [55]. Taken together, we hypothesize that GAs metabolism, high levels of ZR and ABA, and the cytokinin signal pathway may participate in the process of GA-induced axillary meristem formation (Figure 7A).

Besides, a low level of IAA at the leaf axil is required for axillary meristem initiation [51,56]. In this study, a significant decrease in IAA level in stems arose at 8 days after GA_3_ treatment was recorded, which happened after the changes of GAs, ZR and ABA levels, and may promote axillary meristem formation of garlic. Then, an elevation of IAA level at 16 days after GA_3_ treatment and low ZR level at 16 and 32 of GA_3_ treatment could impair axillary meristem (bud) growth and maintain axillary meristem dormancy [4,8,57]. Meanwhile, *AsAUX*, an auxin signal pathway gene, strongly downregulated after 8, 16 and 32 days of GA_3_ treatment, suggesting auxin signal pathway negative regulates axillary meristem formation and dormancy [53,56]. Even though the potential roles of GAs, ZR, IAA and ABA in different stages of GA-induced axillary meristem are suggested in this study (Figure 7A), the effect of application of other hormones (CKs, IAA, ABA and BR) on axillary meristem development of garlic needs to be elucidated, which promotes the application of phytohormones to regulate bud development and improve garlic propagation efficiency in horticultural practice.

Sugars play a critical role in bud formation and development as they are a source of carbon for protein synthesis and provide energy [58,59]. Herein, we supposed a higher content of soluble sugar, glucose and fructose happened after 4 days of GA_3_ treatment to provide energy for axillary meristem initiation, then a low content of soluble sugar at 16 and 32 DAT maintained axillary meristem dormancy [5,21,60]. A large amount of protein participated in the process of axillary meristem formation and dormancy [21,60,61]. Interestingly, low expression of *AsBGLU31* arose at the stage of axillary meristem initiation. Because beta-glucosidase (BGLU) is a rate-limiting enzyme for cellulose hydrolysis, as it converts cellobiose into glucose [62], it is possible that there is a negative feedback loop between glucose and *AsBGLU31*. In addition, sucrose is a modulator of the critical hormone mechanisms controlling shoot branching in *Arabidopsis* [63], *Rosa hybrida* [4] and potato [64]. Our findings agree with previous studies [65,66]—that high sucrose content in buds and stems plays a positive role in axillary meristem dormancy. Invertase, as an hydrolytic enzyme that cleaves sucrose into glucose and fructose, affects plant growth and development [67,68]. Heyer et al. [69] reported overexpressing *apoplastic invertase* (*AI*) in the meristem of *Arabidopsis* enhanced branching of the inflorescence. Nevertheless, high expression of *AsINV* was found in this study, indicating *AsINV* expression was probably induced by high sucrose content to maintain sucrose homeostasis. Whether or not *AsINV* is induced at the stage of axillary meristem release (axillary bud outgrowth) needs further investigation.

Kamenetsky’s group screened more than 1000 DEGs using RNA-sequencing in garlic [32], whereas there were only 159 DEGs (GA_3_ treatment versus water treatment) in this study. Furthermore, DEGs in this study did not involve the key genes regulating axillary meristem, such as *LAS*, *RAX*, *REV* and *WUS*. As we know, shoot apical meristem and axillary bud hide in the basal stem of garlic restricts the observation of axillary meristem development and affects the precision of samples. Therefore, it is suggested the complex and large organ of the sample (1 cm-thick stem piece) or the single time of sample (on the second day after treatment) caused a small number of DEGs. Based on PCA analysis (Figure 6), setting up more sample time points for RNA-seq is possible to obtain a large number of DEGs to draw a clear transcriptomic network of GA-induced axillary meristem formation. Even though there were 159 DEGs in this study, approximately 280,000 unigenes were assembled to provide versatile resources for garlic genome research. In the future, obtaining GA-biosynthesis/signaling mutants in garlic should be done for explaining molecular mechanisms of GA-induced axillary meristem formation.

## 4. Materials and Methods

### 4.1. Plant Materials, GA_3_ Treatment and Growth Conditions

Garlic cultivar G064, widely cultivated in China, was used in this study, which was produced in the experimental station of Northwest A&F University, using regular horticultural practices (sown date: 10 September 2016; harvest date: 25 May 2017). After harvest, the bulbs were put in a well-ventilated and dark room at a temperature of 15–25 °C for three months, then broken into cloves, and uniform cloves were picked for this experiment. The bulb morphological traits of cv. G064 are weight 35–55 g, diameter 3.5–5.5 cm and 11–13 cloves arranged in two whorls.

This study was a pot experiment and included two treatments (GA_3_ treatment and control). At the beginning, 50 uniform cloves were sown in a plastic box (length 60 cm × width 40 cm × depth 20 cm) on 20 September 2017, with six replications. Then, the six plastic boxes were placed in a plastic tunnel of Northwest A&F University, and mean temperature per month and total sunshine hours per month in the plastic tunnel during the garlic growing season were recorded and shown in Appendix A. Lastly, three plastic boxes were randomly selected in which each garlic plant was injected with 2 mL GA_3_ solution (1 mmol L^−1^; Yuanye, Shanghai, China) on 1 November 2017, as the GA_3_ treatment, and garlic plants of another three plastic boxes were injected with distilled water, as control. The method of GA_3_ and water injection was the same as used in our previous study [20]. “Jiahui” substrate (Liaocheng, Shandong Province, China) was used as the growing medium, which contained 20–25% organic matter and 8–10% humic acid. Water, calcium, ammonium nitrate and potassium (Xuzhou, Jiangsu Province, China) were regularly supplemented during the growth season of the garlic. Plant health was controlled by Maneb and Thiometon-based insecticides.

### 4.2. Histology of Axillary Meristem and Axillary Bud

One cm-thick stem pieces of water-treated plants and GA_3_-treated plants were collected at 30 and 90 DAT. Then, the 1 cm-thick stem pieces were immediately put into FAA solution (5 mL formalin, 5 mL acetic acid, 50 mL alcohol and 40 mL distilled water) and samples were stained with 1% safranin for 48 h. In accordance with Lv et al.’s method [60], a 10 μm-thick paraffin longitudinal section was obtained for observing axillary meristem formation under a microscope (BX51 + PD72 + IX71, OLYMPUS, Tokyo, Japan). At 120 and 150 DAT, axillary bud formation was checked in the control group and GA_3_ treatment group by a stereoscopic fluorescence microscope (MZ10F, LEICA, Heidelberg, Germany).

### 4.3. Assessment of Endogenous Plant Hormone Levels

In order to evaluate the changes of endogenous hormone level, sugars content, soluble protein and related gene expression level in stems during this period of GA-induced axillary meristem formation, sampling (3 plants per growing box) in the control group and GA_3_ treatment group was conducted at 2, 4, 8, 16 and 32 DAT. As shown in Appendix A, 1 cm-thick stem was separated from the whole plant and immediately placed into liquid nitrogen, then stored at −80 °C.

The samples (0.5 g) were ground in an ice-cold mortar with 8 ml of 80% (v/v) methanol medium that contained butylated hydroxytoluene (1 mM) as an antioxidant. The extracts were incubated at 4 °C for 4 h and then centrifuged at 500 rpm for 10 min at 4 °C. The supernatants were passed through Chromosep C18 columns (Waters Corporation, Millford, MA, USA). The efflux was collected and dried under nitrogen gas. The residues were then dissolved in 2 mL of 0.01 mM phosphate buffer saline (PBS) containing 0.1% (v/v) Tween 20 and 0.1% (w/v) gelatin (pH 7.5) to determine the endogenous phytohormone levels. As described in previous studies [70,71,72,73], the measurement of zeatin riboside (ZR), indole-3-acetic acid (IAA), abscisic acid (ABA) and gibberellins (GA_3_ and GA_4_) were performed using an indirect enzyme-linked immunosorbent assay (ELISA) technique. The mouse monoclonal antigens and antibodies against ZR, IAA, ABA, GA_3_ and GA_4_ were produced at the Phytohormones Research Institute (China Agriculture University, Beijing, China).

### 4.4. Evaluation of Sugars Content and Soluble Protein Content

Sucrose, glucose and fructose were extracted from mashed 1 cm-thick stem in 10 mL of double distilled water for 30 min at 80 °C. The extracted sample was centrifuged at 12,000 rpm for 10 min at room temperature. The supernatant was filtered through a 0.45 μm nylon inorganic filter for high performance liquid chromatography (HPLC) analyses. The determination method and experiment instruments were the same as research reported previously [74]. The concentrations of sucrose, glucose and fructose were expressed in mg g^−1^ on a fresh weight basis. Soluble sugar content was evaluated by DuBois et al.’s method [75]. Soluble protein content was estimated by the Coomassie Brilliant Blue (CBB) method [76].

### 4.5. Transcriptome Analysis

Based on our pre-experiment and previous studies [21], 1 cm-thick stems of plants in control and GA_3_ treatment were collected at 2 DAT, with three biological replicates. Total RNA was extracted from each sample using *RNAprep* Pure Plant Kit (Tiangen Biotech, Beijing, China). The sequencing library was prepared by random fragmentation of the cDNA sample, followed by 5′ and 3′ adapter ligation. Adapter-ligated fragments were PCR amplified and gel purified. Then, the libraries of mRNA were sequenced on an Illumina Hiseq 2500 platform (Novogene Co. Ltd., Beijing, China; www.novogene.cn) and 150 bp paired-end reads were generated.

Raw reads of fastq format were firstly processed through in-house Perl scripts, which were deposited in the NCBI (National Center for Biotechnology Information, Bethesda, MD, USA) database (SRA accession: PRJNA565115). After adapter sequences and low-quality reads were eliminated, clean reads were used to de novo assemble the garlic transcriptome using the Trinity platform (Broad Institute, Cambridge, MA, USA and Hebrew University of Jerusalem, Jerusalem, Israel) [77]. The assembled unigenes were each searched against five public databases: SWISS-PROT protein database, Gene Ontology (GO) database, NCBI nonredundant protein sequences (NR) database, NCBI nucleotide sequences (NT) database and KEGG ortholog (KO) database. Furthermore, the unigene expression was normalized using the fragments per kilo bases per million reads (FPKM) method described by Mortazavi et al. [78]. The differential gene expression between control and GA_3_ treatment were analyzed using the edgeR software [79] with an FDR of 0.05 and |logFC| ≥ 1 as the threshold. Differentially expressed genes (DEGs) were conducted by GO enrichment analysis and KEGG enrichment analysis using R based on hypergeometric distribution. Significantly enriched GO terms and KEGG pathways were identified, based on the corrected *p*-value (*p* < 0.01 and *p* < 0.05, respectively).

### 4.6. Quantitative Real-Time PCR (qRT-PCR) Validation of Gene Expression Levels

In according to DEGs results, *AsGA20ox*, *AsAUX*, *AsCYP735*, *AsAHK*, *AsBGLU31* and *AsINV* were selected for analysis using RT-qPCR. Gene-specific primers were designed with the Primer Premier 5.0 software (PREMIER Biosoft, San Francisco, CA, USA), as shown in Appendix A. RT-PCR was performed as follows: 95 °C for 3 min; 35 cycles at 95 °C for 30 s; 54–64 °C for 30 s and 72 °C for 20 s; and final extension at 72 °C for 3 min. The RT-qPCR was conducted by the Maxima SYBR Green Master Mix (Thermo Scientific) and a Real-time Quantitative PCR System (iQ5, Bio-Rad, USA). The garlic *actin* gene (*AsACT*) was used as internal reference control to standardize the results [80]. The relative expression data were analyzed using the 2^−∆∆CT^ method. The final values were presented as means of three independent biological trials.

### 4.7. Statistical Analysis

All data were subjected to paired Student’s t-test using SAS 9.2 (SAS Institute, Cary, NC, USA). Principal component analysis (PCA) was also performed by SAS 9.2. All graphs were plotted using the Sigma Plot 10.0 software (Systat Software Inc., San Jose, CA, USA).

## 5. Conclusions

Our results demonstrated that two stages, axillary meristem initiation and dormancy, happened in the period of 0–30 days after GA_3_ treatment, and axillary bud formation occurred twice in GA_3_-treated plants (Figure 7B). Injecting the garlic plant with GA_3_ solution led firstly to the downregulation of *AsGA20ox* and *AsBGLU31* expression, a significant increase in the levels of ZR and ABA and the upregulation of *AsCYP735* and *AsAHK* expression, which may activate axillary meristem initiation. Subsequently, the low level of IAA and high level of soluble sugar (glucose, fructose and sucrose) occurred and probably promoted axillary meristem formation in GA_3_-treated garlic plants. Lastly, low level of ZR and soluble sugar, the downregulation of *AsAUX* expression and high content of sucrose occurred at 16 and 32 days of GA_3_ treatment to inhibit axillary meristem growth and induce axillary meristem dormancy (Figure 7A).

## Figures and Tables

**Figure 1 plants-09-00970-f001:**
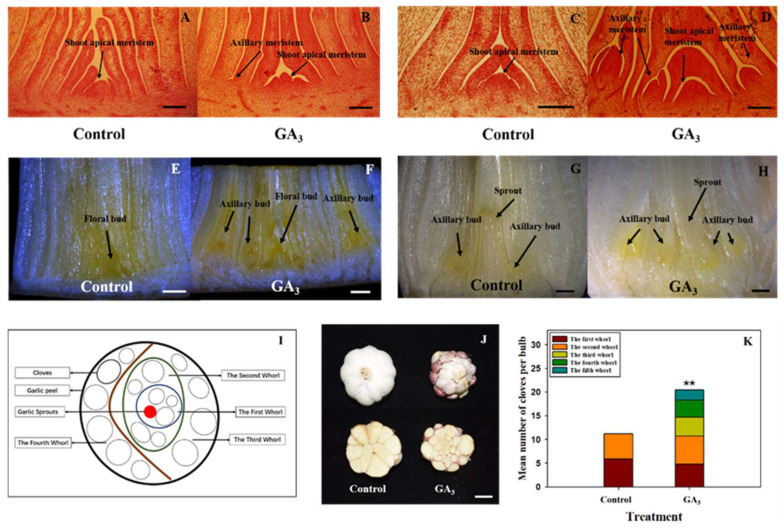
Histological studies on axillary meristem (bud) development in the control group and gibberellins GA_3_ group. Meristem or bud development were observed at 30 (**A**), 90 (**C**), 120 (**E**) and 150 (**G**) days after water treatment. Meristem and bud development were observed at 30 (**B**), 90 (**D**), 120 (**F**) and 150 (**H**) days after GA_3_ treatment. Staining solution was 1% safranin in A, B, C and D. Scale bars in A, B, C and D are 200 μm in length. Scale bars in E and F are 1 mm in length. Scale bars in G and H are 2 mm in length. After harvest, bulb structure was analyzed between control and GA_3_ treatment. (**I**) A model for recording the clove number per bulb and whorl number per bulb. (**J**) Photographs showing representative bulb structure between control and GA_3_ treatment. Scale bar = 1.5 cm. (**K**) Effect of injection of GA_3_ on clove number and whorl number of garlic. Values are mean (*n* = 10) for (**K**). Student’s t-test was used to determine significant differences between the control group and GA_3_ treatment group in (**K**). Significance levels: ** *p* < 0.01.

**Figure 2 plants-09-00970-f002:**
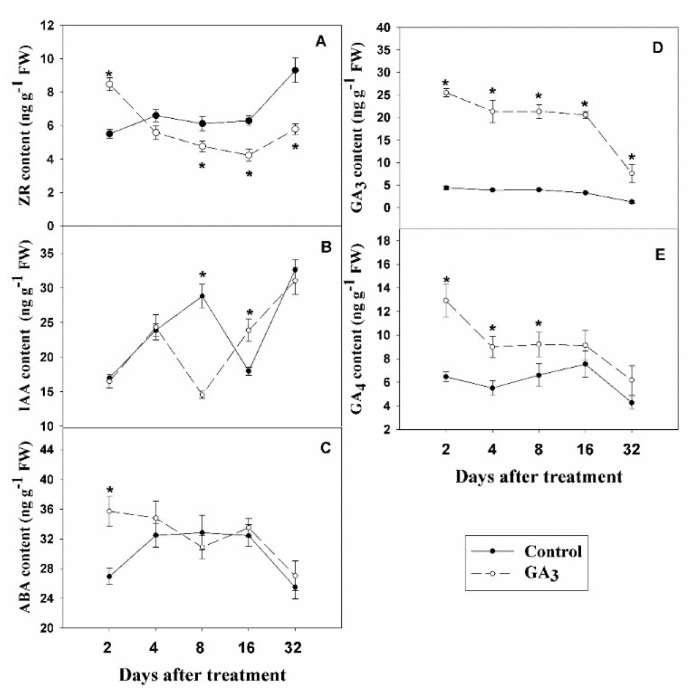
Effect of injecting plants with GA_3_ on the level of zeatin riboside (ZR) (**A**), indole-3-acetic acid (IAA) (**B**), abscisic acid (ABA) (**C**), GA_3_ (**D**) and GA_4_ (**E**) in 1 cm-thick stems of garlic. Values are the means ± SD (*n* = 4) and Student’s t-test was used to determine significant differences at the same period between the control group and GA_3_ treatment group. Significance levels: * *p* < 0.05.

**Figure 3 plants-09-00970-f003:**
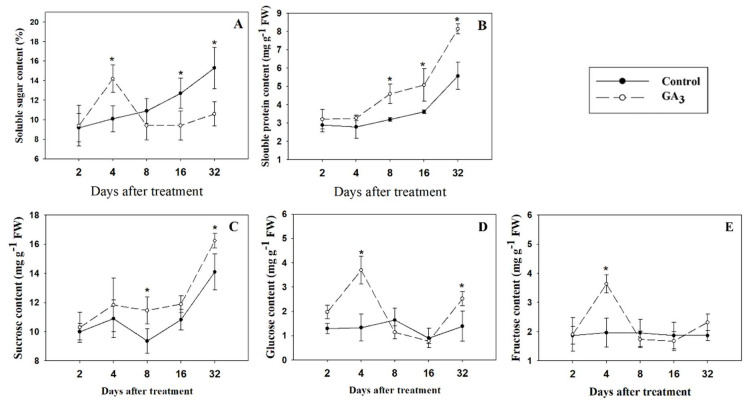
Effect of injecting plants with GA_3_ on sugars content and soluble protein content in 1 cm-thick stems of garlic: (**A**) soluble sugar, (**B**) soluble protein, (**C**) sucrose, (**D**) glucose, and (**E**) fructose. Values are the means ± SD (*n* = 4) and Student’s t-test was used to determine significant difference at the same period between the control group and GA_3_ treatment group. Significance levels: * *p* < 0.05.

**Figure 4 plants-09-00970-f004:**
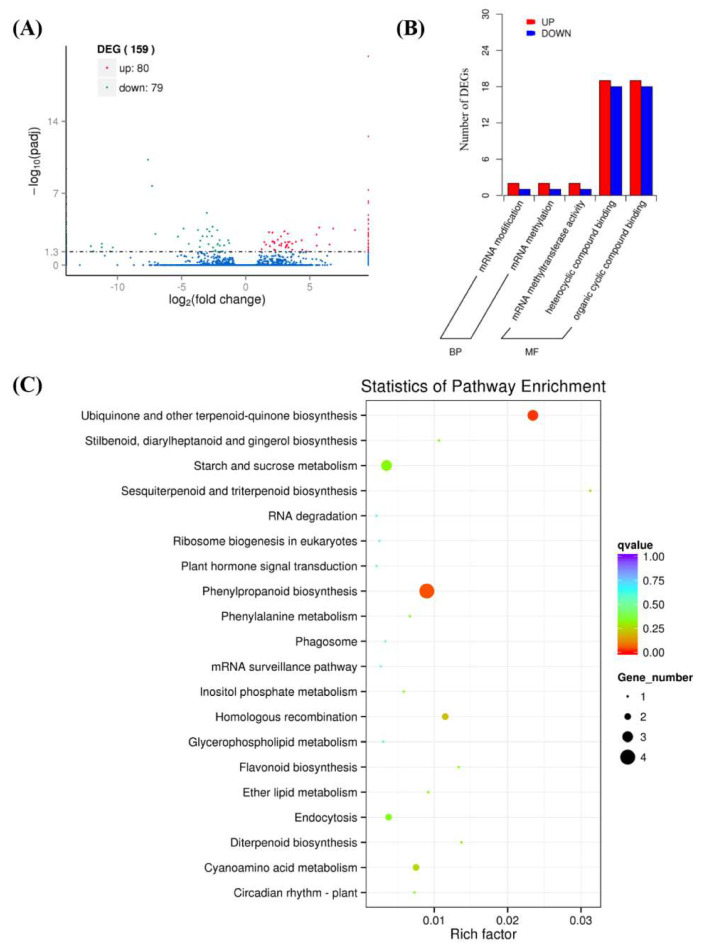
DEGs annotation and enrichment in response to GA_3_ treatment. (**A**) Distribution of the DEGs at 2 days after GA_3_ treatment. The selected level of DEGs was |log2(Fold Change)| > 1. (**B**) GO (Gene Ontology) enrichment of DEGs from GA_3_ treatment versus control. BP: biology process. MF: molecular function. (**C**) KEGG (Kyoto Encyclopedia of Genes and Genomes) enrichment of DEGs from GA_3_ treatment versus control.

**Figure 5 plants-09-00970-f005:**
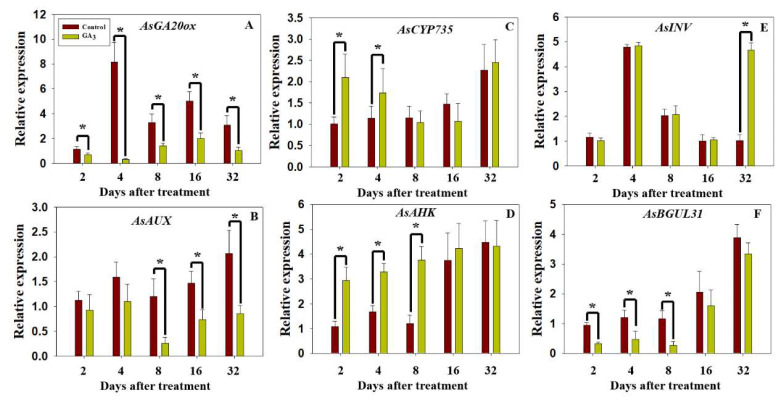
The expression of six key unigenes in stems under GA_3_ treatment versus control (water treatment) at 2, 4, 8, 16 and 32 DAT: (**A**) *AsGA20ox*, (**B**) *AsAUX*, (**C**) *AsCYP735*, (**D**) *AsAHK*, (**E**) *AsINV* and (**F**) *AsBGLU31*. Data represent mean ± SD of three biological replicates, with transcripts normalized to *Allium sativum actin*. DAT, days after treatment. Significance level: * *p* < 0.05.

**Figure 6 plants-09-00970-f006:**
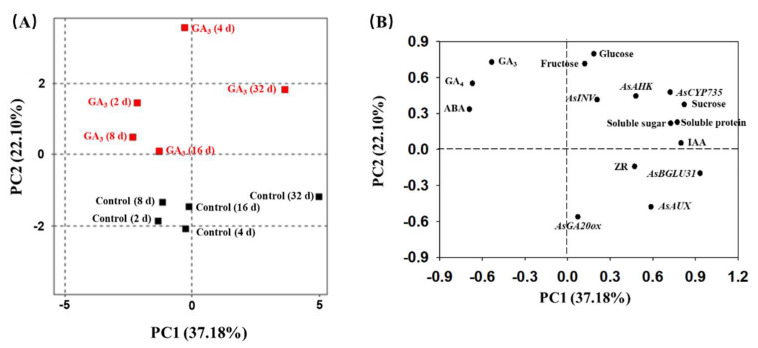
PCA (**A**) and (**B**) loading plots of principal components 1 and 2 obtained from phytohormone level, sugars content, soluble content and expression level of key genes in stems of GA_3_-treated plants and water-treated plants (control) at 2, 4, 8, 16 and 32 DAT.

**Figure 7 plants-09-00970-f007:**
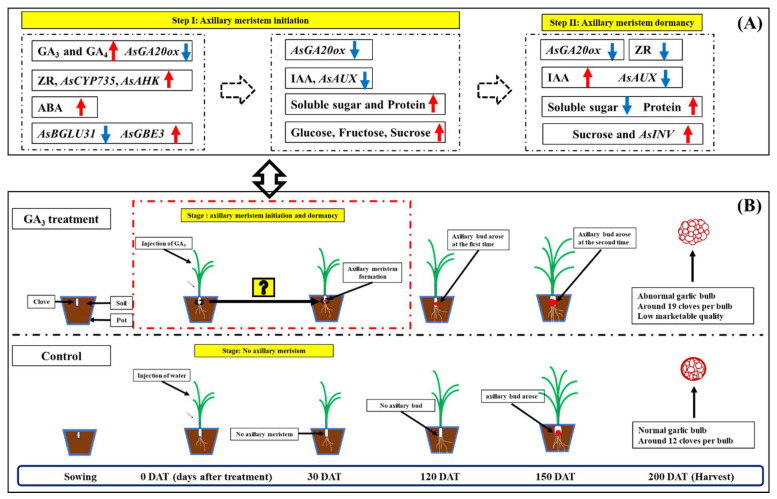
A summary of patterns for GA-induced axillary bud formation in garlic: (**A**) cross-talk of phytohormone and sugar in the process of axillary meristem development of garlic, (**B**) axillary bud (clove) development under water (control) and GA_3_ treatment. Symbols: red arrows indicate upregulation and blue arrows indicate downregulation in (**A**).

**Table 1 plants-09-00970-t001:** Transcriptome data output quality list.

Sample Name	Raw Reads	Clean Reads	Clean Bases	Error Rate (%)	Q20 (%)	Q30 (%)	GC Content (%)	Total Mapped
Control-1	84,276,914	83,119,412	12.47G	0.01	97.69	94.11	43.53	63,037,528 (75.84%)
Control-2	76,333,542	73,699,374	11.05G	0.02	96.86	92.24	43.05	56,814,448 (77.09%)
Control-3	83,928,404	82,410,306	12.36G	0.01	98.1	95.1	42.8	64,086,848 (77.77%)
GA_3_-1	83,633,550	82,546,464	12.38G	0.01	97.78	94.28	43.34	64,036,932 (77.58%)
GA_3_-2	83,869,612	82,222,794	12.33G	0.01	98.14	95.19	42.92	63,589,684 (77.34%)
GA_3_-3	91,727,804	90,117,508	13.52G	0.01	98.16	95.22	43.05	69,904,142 (77.57%)

Raw reads: statistics of the original sequence data. Clean reads: the reads removing low-quality reads. Q20 and Q30: the percentage of bases with *Phred* values > 20 and 30, respectively. GC content: the GC ratio of total base number. Total mapped: the number of reads which can be mapped to the reference genome.

**Table 2 plants-09-00970-t002:** Key unigenes (based on physiological analysis) in the GA_3_ treatment and the corresponding unigenes in the control.

Unigene ID	FPKM		Log2FC	Swissport Annotation
	Control	GA_3_ Treatment		
**Cluster-32430.71391**	0.86 ± 0.09	0.02 ± 0.05	−4.84	Gibberellin 20 oxidase 2 (GA20ox2)
Cluster-32430.95397	2.58 ± 0.84	2.84 ± 0.91	−0.14	Transcription factor PIF4 (PIF4)
Cluster-15446.0	0 ± 0	0.3 ± 0.27	4.91	Cytokinin hydroxylase (CYP735)
Cluster-32430.98625	0.06 ± 0.12	0.44 ± 0.23	2.73	Histidine kinase (AHK)
Cluster-32430.172148	13.41 ± 10.43	6.24 ± 2.01	−1.10	Auxin-induced protein (AUX)
Cluster-32430.118133	6.41 ± 1.07	5.11 ± 1.08	−0.33	Auxin response factor 19 (ARF19)
Cluster-32430.143582	16.9 ± 5.27	26.5 ± 13.78	0.78	6(G)-fructosyltransferase (INV)
**Cluster-32430.190012**	1.4 ± 0.15	0.23 ± 0.36	−2.51	Beta-glucosidase 31 (BGLU31)
**Cluster-32430.4608**	1.47 ± 0.76	0 ± 0	−12.71	Beta-glucosidase 25 (BGLU25)
**Cluster-32430.132754**	0.17 ± 0.15	0.74 ± 0.09	2.21	1,4-alpha-glucan-branching enzyme 3 (GBE3)
Cluster-49254.0	0.43 ± 0.12	0.32 ± 0.11	−0.42	WUSCHEL-related homeobox 3 (WOX3)
Cluster-32430.173042	5.86 ± 2.84	4.07 ± 1.12	−0.53	WUSCHEL-related homeobox 10 (WOX10)

FPKM represents unigene expression level that is normalized by FPKM (fragments per kilobase per million mapped reads) approach. The unigenes in bold are differentiated expression genes (DEGs) between control and GA_3_ treatment samples at 2 days after treatment.

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
