# Peer review of "Histological, Physiological and Transcriptomic Analysis Reveal Gibberellin-Induced Axillary Meristem Formation in Garlic (Allium sativum)"

_plants, 2020, doi:10.3390/plants9080970_

Round 1

Reviewer 1 Report

This article described that injection of GA3 enhanced axillary bud development and increased clove numbers. The authors showed morphological changes, sugars/protein content, genes alternation after GA3 treatment. It is more solid if authors can treat inhibitor of GA3, such as PBZ, inhibit clove number, that will provide more strong evidence to demonstrate the essential role of GA-induced axillary meristem formation in garlic.  

Need to revise

  1. Line 75, Based on our previous studies [20,21], please summarize some major outputs.
  2. Line 141, transcriptome analysis using RNA after GA3 treatment at 2 DAT … any reason. Sugar/ protein contents were found significant different at 2 DAT onwards. Whether that caused low DEGs (159 DEGs, Line 299) were found in this study?
  3. Figure 1, please mentioned tissues were stained with what buffers? 
  4. Table 2, please add fold change and standard deviation of three-bioreplicates (?).
  5. Please italic “Allium sativum” and genes in all text, including lines 183, 184 etc.
  6. Line 246, Symbols “red arrow” indicate up-regulated genes, and “blue arrow” indicated down-regulated genes in (A), should include biochemical and plant hormones contents.   
  7. Line 324, garlic plants were injected with 1 mmol L-1 GA3 solution…Please mention how many mL was injected per plant?  
  8. Line 408, I could not see Supplementary Materials: The following are available online at mdpi.com/xxx/s1.... Can’t open in the website. Please attached the data in a pdf.  

Reviewer 2 Report

In this article, the authors used a multidisciplinary approach to study the role of GA3 in promoting axillary bud formation. The analysis was performed using a combination of molecular, physiological, and chemical approaches, as well as described alongside the paper. The paper is interesting, and It seems adequate for publication in this journal.

The main problem of this work is a possible case of plagiarism with an article published by the same research group: "Response of axillary bud development in garlic (Allium sativum L.)
to seed cloves soaked in gibberellic acid (GA3) solution", published in Journal of Integrative Agriculture 2020, 19(4): 1044–1054. This article was not reported in the reference section.
Figure 8 reported in the already published article is the same as Fig. 7 reported in the present article. Also, Figure 1 is similar to Fig. 3 (panel A) reported in the already published article.
Furthermore, some sentences are reported in both the articles: e.g. line 317-318 (data related to bulb morphological traits) and third row in section "Plant materials and GA3 concentration" in the already published article.

Other comments

English language. The overall quality is quite good, but I suggest an English revision by a mother tongue. Some errors are present, mainly in the Discussion and Results section (some of them are reported below).

Introduction. This section is lacking some important references, as, for instance:
Plant physiology group at the University of Pisa (authors Mariotti, Pucciarelli, Ceccarelli).
Shinjiro Yamaguchi (Tohoku University).
I strongly suggest improving our introduction by adding more articles related to the state-of-the-art in gibberellins study.

qPCR data (Figure 5) were used to confirm RNAseq data on some selected genes (Table). Have you tried to correlate RNAseq and qPCR for these genes? (e.g. through a Pearson correlation graph?) Which was the r square correlation value?

The relative expression values reported in Figure 5 do not present any measuring unit (log10?)? Please, check it.

PCA was performed to achieve a direct comparison of data coming from different analyses, but the information reported about PCA computing is poor. Did the authors normalize data prior to statistical analysis? How did they compare data like those related to sugar and hormone content that usually present different scale of values?

Line 144-147; This sentence is questionable since many other factors (e.g. quality of reference genome) can influence the mapping rate. I suggest to modify or to remove this sentence.

Line 149. Differentially instead of differential
Line 150. These acronyms (GO and Kegg) never appear before. Please, insert their meaning here.
Line 196. Was instead of were.

Figure 6 caption. Components instead of component

Line 237. I suggest modifying this sentence; "30 and 60 DAT might have been in winter..."

Line 260. "Cytokinin levels" instead of "Cytokinin level".

Line 268. I suggest changing "In addition" with "Besides, a low level"
Line 269. Was instead of were
Line 287. Converts instead of Convert.
Line 295. Homeostasis instead of homostasis.
Line 299. Were instead of was.

Line 307-310. This sentence seems not appropriate for ending the discussion section. Please, check it.

Line 316. I suggest changing the term "sunless" with "dark".
Line 359. Rpm instead of rmp

The conclusion is too short. I suggest improving this section to better highlight the results of this work.

Round 2

Reviewer 2 Report

I carefully read all the author's comments, as well as the text modifications. Most all the issues were fixed, but I think that too many parts of the present paper remain too similar to the already published one, ()" The pictures can't be too similar due to copyright laws, to avoid any further problem between publishers. This is the main reason leading my "major revision" choice: I was hoping that the authors would change these two pictures, but they didn't'. Also, the author explanations t "... we used the same research method both this paper and this manuscript, which caused the similar pictures in this paper and manuscript..." is not persuasive, since it can be read as "the two works derived from a single experiment, where results were split to get two publications.", which is not optimal in terms of publication ethics.
Considering that, I have to reluctantly refuse the work for publication on Plants. Despite that, I strongly suggest authors resubmit the paper once all they will be fixed all the article's issues.
